# A Comprehensive Neuropsychological Study of Familial Hypercholesterolemia and Its Relationship with Psychosocial Functioning: A Biopsychosocial Approach

**DOI:** 10.3390/brainsci12091127

**Published:** 2022-08-25

**Authors:** Moon Fai Chan, Aishwarya Ganesh, Sangeetha Mahadevan, Siham Al Shamli, Khalid Al-Waili, Suad Al-Mukhaini, Khalid Al-Rasadi, Samir Al-Adawi

**Affiliations:** 1Department of Family Medicine and Public Health, College of Medicine & Health Sciences, Sultan Qaboos University, Al-Khoud, Muscat 123, Oman; 2Department of Behavioural Medicine, College of Medicine & Health Sciences, Sultan Qaboos University, Al-Khoud, Muscat 123, Oman; 3Department of Psychiatry, Ibri Hospital, Ibri 516, Oman; 4Department of Clinical Biochemistry, Sultan Qaboos University Hospital, Al-Khoud, Muscat 123, Oman; 5Directorate of Nursing, Sultan Qaboos University Hospital, Al-Khoud, Muscat 123, Oman; 6Medical Research Center, College of Medicine & Health Sciences, Sultan Qaboos University, Al-Khoud, Muscat 123, Oman

**Keywords:** familial hypercholesterolemia, psychosocial functioning, cognition, biopsychosocial model, Oman

## Abstract

**BACKGROUND**: Over the past few years, there has been an increasing interest in viewing the diagnosis of familial hypercholesterolemia (FH) through the lens of the biopsychosocial model. However, other than a few epidemiological surveys, there is a dearth of studies from emerging economies that have examined FH using the biological, psychological, and socio-environmental facets of the aforementioned model. **AIM**. The three aims of the current study were as follows: (i) to examine the psychosocial status among patients with genetically confirmed FH, (ii) to compare their intellectual capacity and cognitive outcomes with a reference group, and (iii) to examine the relationship between health literacy and cognitive functioning. **METHOD**: Consecutive FH patients referred to the lipid clinic at a tertiary care center for an expert opinion were recruited into this study conducted from September 2019 to March 2020. Information regarding psychosocial functioning, health literacy, quality of life, and affective ranges was surveyed. Indices of current reasoning ability and cognition (attention and concentration, memory, and executive functioning) were compared with a socio-demographically-matched reference group. The current hypothesis also explored the impact of FH on health literacy and cognition. **RESULT**: A total of 70 participants out of 106 (response rate: 66.0%) initially agreed to participate. However, 18 out of 70 dropped out of the study, yielding a final total of 52 FH patients. With 27 (51.9%) males and 25 (48.1%) females, the mean participant age stood at 37.2 years (SD = 9.2), ranging from 21 to 52 years of age. In the psychosocial data, thirty-two percent (n = 17) of them had anxiety (HADS ≥ 8), and twenty-five percent (n = 13) had depressive symptoms (HADS ≥ 8). The performance of the FH patients was significantly impaired compared to the control group on the indices of current reasoning ability and all domains of cognitive functioning. In the univariate analysis conducted to compare cognitive functioning with health literacy status, only indices of attention and concentration emerged as being significant. **CONCLUSION**: The current study indicates that the FH population is marked with impediments in biopsychosocial functioning, including indices tapping into the integrity of health literacy, quality of life, affective ranges, and higher functioning such as cognition and current reasoning ability when compared with a socio-demographically-matched reference group. The present results support the hypothesis that chronic diseases vis-à-vis the sequelae of coronary artery disease can potentially impede biopsychosocial functioning.

## 1. Introduction

Familial hypercholesterolemia (FH) is a medical condition characterized by elevated low-density lipoprotein (LDL) or low-density lipoprotein cholesterol (LDL-C). In the extant literature, LDL-C has been labeled as “bad cholesterol” since it has generally been associated with a heightened risk of early onset of coronary artery disease if not sufficiently treated. It has been estimated that more than 25 million people are marked with FH worldwide [1,2]. According to the ‘Global Familial Hypercholesterolemia Community’, if the presence of FH is not detected early and prompt lifestyle changes and medical mitigation are not undertaken, it often leads to premature morbidity and mortality due to atherosclerotic cardiovascular disease [1]. In context of the biopsychosocial model, there is plenty of literature on the genetic and biological mechanisms of FH. However, little has been forthcoming on the psychosocial aspects of this condition.

On the physical front, hypercholesterolemia has been recognized for decades as a potential risk factor for cardiovascular disease development. Emerging evidence has also indicated a strong association between cholesterol dyshomeostasis and cognitive impairment [3]. In support of this view, many preclinical studies suggest that mechanisms that are involved in the expression of FH tend to disrupt the integrity of the blood–brain barrier. According to the animal model of FH, this condition, in turn, compromised critical brain regions that are involved in cognition [4,5]. Other preclinical studies have also garnered support for this link between FH and the integrity of cognition [5,6].

Evidence from clinical literature has also suggested that people with FH are likely to show cognitive decline. In a Spanish population, Zambón et al. [7] compared the performance on indices of cognitive decline among FH patients (N = 47) and matched controls (N = 70), reporting that FH patients tend to exhibit a high rate of cognitive impairment compared with those belonging to the non-FH clinical population. Another study from Spain explored cognitive status among young people with FH. This study appears to suggest that even a young FH population tends to exhibit cognitive decline [3]. Suárez Bagnasco [8] reviewed articles appearing in various medical and psychological search engines from 1980 until March 2017. The search accrued five studies, suggesting that cognitive impairment is common among FH patients between 18 and 40 years old. This suggests that cognitive impairment in FH is not an artifact of aging or longevity like in other types of dementia, where age is a critical component among the majority of people with cognitive decline.

The question then arises whether pharmaceutical agents that are typically used to modulate cognition could improve cognitive decline in FH patients as well. This issue was explored by Lopes et al. [9] in preclinical literature, whereby cholinesterase inhibitors such as Donepezil could be used to reverse cognitive decline and prevent its progression in an animal model of FH. Additionally, evidence from studies on statin therapy has shown that the compounds that have been established to reduce LDL and confer protection against cardiovascular sequelae of FH have been suggested to reduce cognitive decline as well [10,11,12].

As neuropsychological impairment is common among people with FH, there is the hypothesis that FH patients will exhibit functional and structural changes in the brain. Todate et al. [13] have examined the effects of hypercholesterolemia on cerebral small vessel disease in individuals with elevated LDL-C using high-resolution brain magnetic resonance imaging (MRI). This study revealed a significantly higher prevalence of cerebral damage (e.g., lacunar infarction, deep white matter hyperintensities, micro-bleeding, brain atrophy, etc.) among FH patients as compared to controls. 

The aforementioned temporal relationship between FH and neuropsychological impairment implies that people with FH (PwFH) are likely to have problems with attention and concentration, learning and remembering, executive functioning, as well as other phenotypical presentations of cognitive decline. Therefore, FH vis-à-vis cognitive impairment could potentially also impact psychosocial variables such as quality of life. While there is a dearth of studies on the interplay between cognitive impairment and quality of life among PwFH, thereby necessitating such an undertaking, there are a plethora of studies that have examined the quality of life among PwFH. Mata et al. [14] have reported a study in which they accrued a cohort of Spanish participants (n = 1947) diagnosed with FH (n = 1321) and those without FH (n = 626). A similar investigation was conducted among Swedish participants with FH [15], with quality of life tapped into by the 12-Item Short-Form Health Survey questionnaire. These studies concluded that quality of life did not differ between the two cohorts. Mortensen et al. [16] have conducted qualitative studies on the quality of life among FH patients, which suggest that a better quality of life strongly hinges on the utilization of best existing practices for the treatment of FH, as those that met treatment goals were more likely to report adequate QoL compared to those who failed to adhere to the prevention and mitigation protocol. Akioyamen et al. [17] conducted a systematic review and meta-analysis up to 1 January 2018. Among 10 studies that fulfilled the inclusion criteria, psychological domains related to the quality of life (QoL) were significantly elevated only among people with FH. Parallel to QoL, this review also examined the presence of poor mental health outcomes among people with FH [8], concluding that anxiety symptoms might be one of the sequelae of FH. However, so far, there is no evidence of an increased risk of depressive symptoms. Further examination of the poor mental health outcomes in FH is warranted, since the presence of anxiety and depression tends to impinge on cognitive status [18].

Recently studies exploring health literacy (HL) among people with chronic diseases have emerged [19]. The concept of HL implies that afflicted individuals are capable of processing and understanding the healthcare factors affecting their utilization and the treatment accrued [20]. Existing literature has also suggested that higher integrity of HL is often tied with positive prognostic indicators and, conversely, inadequate HL often results in poor utilization of existing healthcare services, poor adherence to the prescribed medication, and a sub-par lifestyle [21,22,23,24,25,26]. While the magnitude affects 1 in 200 to 250 people around the world [1,2], to date, studies on cognitive status, quality of life, mental health status, and health literacy appear to be emanating largely from Western countries. Although some studies have explored HL among FH patients and have documented growing HL in FH, to date, little has been forthcoming from Arabian Gulf countries on the same. This is an important tenet in existing data trends, as sociological and anthropological studies from these countries have suggested that illness behavior in this region appears to be characteristically unique [27,28]. In cultures from this region, afflicted individuals are more likely to adopt the ‘sick person’ role that is congruent with the interdependence of a collective society [29]. Self-initiative and individualistic behavior of the afflicted individual may be curtailed as their family takes over the burden of care for them. It remains to be seen how people with FH fare on the various indices of HL in these regions.

Given the aforementioned background, the three primary aims of the current study were as follows: (i) to examine psychosocial status among genetically confirmed FH patients in Oman; (ii) to compare the intellectual capacity and cognitive outcomes with a socio-demographically-matched reference group; and (iii) to examine the relationship between health literacy and cognitive functioning. Since it would also be theoretically interesting to see if studies would address the hypothesis that FH—which has been widely documented to transcend race and ethnicity [30]—is also marked by psychosocial impediments that have been reported largely in developed countries, this study also aims to fill this gap in the literature. Assuming that the biopsychosocial model deciphers complex mechanisms, this paper also aims to employ an in-depth idiographic approach that captures intra-individual variations in psychosocial and cognitive variables, as these tend to be heterogeneous across FH populations.

## 2. Materials and Methods

### 2.1. Setting

The healthcare setting in Oman is a universal free system for Omanis and is compartmentalized into primary, secondary, and tertiary care. SQUH is a tertiary hospital and a catchment area for national referrals, with its lipid clinic considered a referral center for FH patients across the country. There are two such comprehensive centers in the country; the present study constitutes a single-center study. Currently, most suspected FH cases (individuals with a high probability of having FH according to DLCN criteria and those who did not respond to low-intensity statin therapy) are referred to the SQUH lipid clinic for genetic confirmation and advanced management [31] (Figure 1). Medium-intensity statin therapy to high-intensity statin therapy, Statin + ezitimibe combination, or lipopheresis, are parts of a comprehensive management protocol for patients admitted here with FH [31]. Since invasive lipid apheresis has the potential to be strenuous, the measures were all administered prior to treatment in order to avoid the adverse effect of the procedure.

### 2.2. Diagnosis of Familial Hypercholesterolemia

The methods of FH diagnosis utilized at the present tertiary care center have been described in a study by Al-Waili et al. [31]. To summarize, patients with high levels of low-density lipoprotein cholesterol (LDL-C) of >4.9 mmol/L (>189 mg/dL) were stratified according to the Dutch Lipid Clinic Network (DLCN) criteria [32]. The diagnosis of FH using the DCLN score was derived from different domains, including clinical, family history, physical examination, baseline cholesterol levels, and FH genetic testing. A case of ‘probable/definite’ FH was considered when the DLCN score was 6 or higher, and ‘possible’ FH was considered when the DLCN score was between 3 and 5. Those with DLCN scores of <3 were classified as ‘unlikely’ to have FH.

All patients were also genetically confirmed to have FH. Genomic DNAs were extracted from whole blood using Qiagen mini kit (QIAamp DNA Mini). Ampli-Seq technology on the Ion Proton platform (Thermo Fisher Scientific, Inc., Waltham, MA, USA) was used to sequence the LDL receptor (LDLR), the proprotein convertase subtilisin/kexin type 9 (PCSK9) gene, the gene of apolipoprotein B (ApoB) or rare mutations in LDL receptor adapter protein 1 (LDLRAP1) gene.

Sequencing data were processed by the Torrent Suite and reads were aligned to the hg19 reference genome. Variant call files were then generated using Torrent Variant Caller plugins. Variant annotation was performed using ANNOVAR and variants were linked to ExAC and Greater Middle East-Arabian Peninsula (GME-AP) databases for allele frequencies. The effect of amino acid changes was predicted by LRT, CADD, and MutationTaster. Pathogenic variants were identified from allele frequency of <1% or novel, coverage depth >30, and the damaging effect from at least two of the three prediction algorithms.

### 2.3. Inclusion/Exclusion Criteria

Patients with a history of uncontrolled diabetes, hypothyroidism, acquired brain injury, history of overt neurological events, history of infection that is known to affect cognition, pervasive and persistent mental health problems, intellectual disability, or cognitive decline, as defined by the Montreal Cognitive Assessment [33] (that is, a score of 22/30 or less), were excluded from the study. This cut-off is relevant to further enhance the homogeneity of the cohort. 

The criteria for the present cut-off were based on normative data from the Arabian Gulf [34]. All psychometric evaluations for the present study were conducted and performed by a qualified neuropsychologist.

### 2.4. Control Group

Oman has yet to develop normative data for neuropsychological batteries [35]. In line with the methodology of studies that have examined neuropsychological status among people with Beta-Thalassemia Major [36,37], healthy volunteers were invited to participate in this study as a comparative group. Healthy controls matched for the socio-demographic background were recruited from amongst the staff of Sultan Qaboos University (n = 43). Inclusion criteria for the healthy controls included those with a clean bill of health and no evidence of a persistent and pervasive history of medical, psychiatric, or neurological complications that resulted in seeking medical attention. This was verbally corroborated by all healthy volunteers.

### 2.5. Ethical Consideration

The study was conducted according to the guidelines of the Declaration of Helsinki and was approved by Sultan Qaboos University’s Medical Research Ethics Committee (MREC #1788). All patients were required to sign informed consent forms before participating in the study.

### 2.6. Outcome Measures

#### 2.6.1. Psychosocial Functioning

Health Literacy:

The 3-item Health Literacy questionnaire was developed by Chew, Bradley, and Boyko [19] by administering the same to patients preparing to undergo ambulatory surgery in Seattle, USA to tap into HL. The three items were related to problems associated with learning (‘How confident are you in filling out medical forms by yourself?’), confidence with medical forms (‘How often do you have someone help you read hospital materials?’), and help required for reading medical literature (‘How often do you have problems learning about your medical condition because of difficulty understanding written information?’). The questionnaire is based on a 5-point Likert scale ranging from ‘never’/’extremely’ (1) to ‘Always’ (5). The scores typically range from 3 to 15; participants scoring ≥3 on each item and greater than 9 on the total score, were classified as reporting adequate HL. In contrast, a score ≤3 on each item and ≤9 on the total score were classified as reporting inadequate HL [22,23]. This scale has been previously utilized to assess HL among those with FH from Australia, Brazil, China, Hong Kong, Malaysia, Taiwan, and the UK [38].

Quality of Life:

The WHO QOL-BREF is a cross-culturally comparable instrument that measures the QoL, and it is publicly available. The Arabic version of the WHOQOL-BREF contains 26 items collapsed into four constructs—physical health, psychological health, social relationship, and family environment—and has been extensively employed among Arabic-speaking populations, including those from Oman [39,40]. The WHO QOL-BREF has 4 domains (physical, psychological, social relations, and environment) and two items evaluate the overall quality of life and general health. Information is reported on a five-point Likert scale (‘how much’, ‘how completely’, how often, ‘how good’, or ‘how satisfied’), with higher scores indicating better quality of life. For theoretical reasons, the established cut-off of ≤60 will be used to differentiate those who have ‘adequate’ (>60) or ‘inadequate’ (≤60) QoL [41].

#### 2.6.2. Intellectual Ability and Cognition

Raven’s Progressive Matrices:

Raven’s Progressive Matrices (RPM) was employed to tap into nonverbal reasoning ability [42], a measure orthogonal to linguistic and scholastic skills. It is comprised of 60 items grouped into five sets; each item consists of a pattern with one part removed and six and eight pictured inserts, each of which contains the appropriate missing part. The participants were required to point to what they perceived to be the correct insert for each pattern. This test measures reasoning ability or the “meaning-making” component of Spearman’s fluid intelligence often referred to as ‘general intelligence’. As detailed in the RPM manual, the raw scores (out of 60) are then tabulated into percentile scores based on chronological age. Thus, the present score constitutes a percentile score. The following sections will briefly explore the three major dimensions of intellectual ability.

Attention and Concentration:

Digit Span derived from the Wechsler Adult Intelligence Scale [43] was used to tap into the variations in attention and concentration across. Both versions of the Digit Span test—igit Span Forward and Digit Span Backward—were used and were then scored separately. There is evidence to suggest that these two versions measure two different domains [44].

Learning and Remembering—Memory:

Participants’ ability to learn and remember was gauged using the California Verbal Learning Test (CVLT), which consisted of 16 shopping list items. This test is designed for tapping into verbal learning and memory strengths and deficits [45]. For the present study, 16 items relating to ‘Long Delay Free Recall’ were utilized as a part of the questionnaire. 

Executive functioning:

Executive functioning constitutes an amalgamation of complex cognitive processes, including planning, working memory, and domains that reflect the temporal organization of behavior and self-regulation. The present study employed two executive functioning measures, namely, the Verbal Fluency Test/Controlled Oral Word Association Test (COWAT) and Trail Making Test [46,47].

Verbal Fluency Test:

The integrity of verbal fluency or phonological fluency was solicited using the COWAT. The participant is required to generate as many different words as possible starting with each of the three specific letters. As previously ascertained to have heuristic value in the Arabic language, the letters were taa, raa, and waaw [48], with 30 s allowed per letter. The total score for COWAT was the total number of different acceptable words produced across three 90-s periods.

Trail Making Test:

There is vast literature suggesting that the Trail Making Test solicits the integrity of executive functioning and psychomotor speed [49]. The Trail Making Test has two versions of the test, Form A and Form B. The present study utilized Form B, whereby the examinee was asked to draw a line to connect, in alternating sequence, digits (1 through 12) and letters (A through L). The performance was scored in terms of seconds. 

#### 2.6.3. Affective Range

Affective ranges were solicited using the Hospital Anxiety and Depression scale (HADS) by Snaith and Zigmond [50]. The HADS is a 14-item symptom checklist designed to tap into the symptoms of anxiety and depression, with 7 items for each. For the present study, the Arabic version of the HADS was used. A cut-off ≥ 8 is considered to constitute case-ness for either depression or anxiety [51].

### 2.7. Statistical Analysis

Descriptive statistics (e.g., mean, median, standard deviation, frequency, percentage) were used to describe the profiles of the study samples. Univariate relationships between groups (FH vs. control) on intellectual capacity and cognitive outcomes were tested by an independent t-test. The Mann–Whitney U test further assessed the performance of FH patients and controls with regard to their health literacy status and cognitive outcomes. All tests of statistical significance were set at *p* < 0.05, and data analysis was performed using IBM SPSS Statistics, v23 (SPSS, Chicago, IL, USA).

## 3. Results

This section may be divided into subheadings. It should provide a concise and precise description of the experimental results, their interpretation, as well as the experimental conclusions that can be drawn.

### 3.1. Sample Characteristics

A total of 70 participants (response rate is 66.0% = 70/106) agreed to participate. However, 18 out of 70 refused to continue, so only 52 FH had completed the study, as shown in Figure 1. Among these 52 patients, 4 were classified as being homozygous, while the rest were all heterozygous. Of those that refused the study, their characteristics did not differ from the cohort that completed the study.

Details of demographic and clinical profiles of the FH and healthy control groups are shown in Table 1. In the FH group, there were 27 (51.9%) males and 25 (48.1%) females. The mean age was 37.2 (SD = 9.2), ranging from 21 to 52 years. The majority of the samples (82.7%, n = 43) were married, their social status in terms of family income was middle class (78.8%, n = 41), and half of them (50.0%, n = 26) had received a university education. In the control group, there were 17 (39.5%) males and 26 (60.5%) females. The mean age was 39.1 (SD = 5.8), ranging from 29 to 56 years. The majority of their social status in terms of family income was middle class (62.8%, n = 27), and half of them (55.8%, n = 24) had received a university education and were single (58.1%, n = 25). There were no significant differences between the two groups on all demographic variables except marital status (*p* < 0.001).

### 3.2. Aim 1: To Examine the Psychosocial Status of Patients with FH

As per psychosocial data, only fifteen percent (n = 8) of them had cardiovascular disease, and more than sixty-five percent of them (n = 34) had adequate health literacy status (Table 1). 

In Table 2, thirty-two percent (n = 17) of them had anxiety (HADS ≥ 8), and 25% (n = 13) had depressive symptoms (HADS ≥ 8). The mean ± SD on current reasoning ability, attention and concentration, long delay-free recall, and two indices of executive functioning (Verbal Fluence and Trail Making Test) scores were 31.8 ± 10.8, 16.8 ± 2.6, 8.9 ± 2.5, 16.0 ± 4.7, and 217.7.2 ± 70.5, respectively.

### 3.3. Aim 2: To Compare the Intellectual Capacity and Cognitive Outcomes with the Reference Group

Table 2 shows the univariate analysis of the intellectual capacity, cognitive outcomes, and affective ranges of people with FH versus a healthy control group. 

Those in the FH group showed a lower index of current reasoning ability (Raven’s Progressive Matrices: t = 2.37, *p* = 0.02), attention and concentration (Digit Span: t = 3.94, *p* < 0.001), long delay free recall (California Verbal Learning Test: t = 4.52, *p* < 0.001), executive functioning (Verbal Fluency: t = 4.23, *p* < 0.001), and reduced anxiety (χ^2^ = 4.18, *p* = 0.041). However, the FH patients generally scored higher than the healthy control group on another index of executive functioning: the Trail Making Test (t = 11.45, *p*< 0.001). 

### 3.4. Aim 2: To Compare the Intellectual Capacity and Cognitive Outcomes with the Reference Group

Table 3 shows the univariate analysis of cognitive outcomes wherein patients with FH were examined for their HL status (Inadequate HL vs. Adequate HL). FH patients with inadequate HL showed significantly lower scores on indices of attention and concentration than the adequate HL group (Digit Span: U = 186.00, *p* = 0.017). However, no significant differences were found between both HL groups in their current reasoning ability (Raven’s Progressive Matrices: U = 244.50, *p* = 0.196), long delay-free recall (California Verbal Learning Test: U = 271.00, *p* = 0.495), executive functioning (Verbal Fluency: U = 252.50, *p* = 0.301), and an index of executive functioning, the Trail Making Test (U = 277.50, *p* = 0.818).

## 4. Discussion

Familial hypercholesterolemia (FH) has been recognized to be “one of the most common inherited monogenic disorders in the general population” [52] (p. 965). Surveys have suggested that 1 in 313 people are afflicted with FH and that such rates transcend culture, ethnicity, and geography [1,53]. While FH was initially perceived to be a purely organic medical condition with critical genetic involvement [54,55], FH has recently been found to have all the hallmarks of a disorder with biopsychosocial characteristics [56]. This has spurred nascent research on the psychosocial aspects of FH [57,58,59]. However, the majority of these studies emanate from industrialized countries of the Global North. To date, besides surveys that have examined the prevalence of FH and its risk factors, there is a dearth of studies that have examined psychosocial factors as the associated variable. The disease process of FH has been shown to compromise the integrity of critical brain areas involved in cognition and emotion which, in turn, has negative repercussions on psychosocial functioning due to the higher risk of CVD [8]. The advent of the use of the biopsychosocial model under scrutiny here postulates that FH is a multimorbid condition and potentially life-limiting disease which might parallel other chronic diseases in terms of triggering long-term disability and dependency [60,61].

Given this background, the current study embarked on examining three major themes. The first was the psychosocial status among genetically confirmed patients with FH, the second was to compare the intellectual capacity and cognitive outcomes of FH patients with the reference group, and the third was to examine the relationship between health literacy and cognitive functioning among FH patients. This study has the advantage of employing a depth of information encompassing the ‘top-down’ and ‘bottom-up’ aspects of the participants, from health literacy to cognition. An in-depth approach of this sort is likely to fulfill the criteria of an idiographic study, an undertaking that is contrasted with the nomothetic approach [62].

The major psychosocial variable explored as part of this study was Quality of life (QoL). The indices of QoL most affected in the present cohort were the physical and environmental domains, followed by domains of social relationships and general health. In addition to QoL, mood symptoms or affective ranges were also explored. In the present study, using the HADS assessment, 32.7% and 25% of the sample with FH from the present study were marked with anxiety and depression, respectively. As of now, the presently observed figures appear to be in higher ranges compared to previous studies, though such comparison is marred by the heterogeneity of the instruments used [63,64,65]. More studies examining mood symptoms in the FH population are therefore warranted. Mood symptoms have been suggested to rank as the second leading cause of global dependence and disability and the presence of these symptoms has also been shown to have a temporal relationship with the development of life-limiting diseases including coronary artery disease which has been intimately linked with FH [66,67]. Studies have suggested that life-limiting diseases tend to contribute significantly to the psychological burden that manifests, mood symptoms being one part of this [17]. Despite its wide prevalence, with a few exceptions, the presence of mood symptoms in the FH population has not received the required attention [64,65].

While clinical and epidemiological surveys have been conducted among patients with FH in emerging economies, to date, there is a dearth of studies on the effects of the disease process on higher functioning. The diminution of higher functioning has been attributed to the effects of elevated levels of cholesterol and the lack or dysfunction of LDL receptors, contributing to the development of cognitive impairment among patients with FH [3].

In the present study, the outcome measures that were employed lack local normative data as a reference, except for the indices of current reasoning ability—Raven Progressive Matrices. Percentile scores on indices of reasoning ability among FH patients were within the average range of intellectual ability (percentile = 31.8 ± 10.8) [68]. However, the reference group scored in the elevated percentiles. For the measurement tapping into cognitive functioning, the study compared the score of the patients with FH against those of the reference group. The two groups were statistically different on indices of attention and concentration, learning and remembering, as well as two measures of executive functioning, solicited from the Verbal Fluency Test and the Trail Making Test. Due to divergent neuropsychological measures that are often employed to tap into the cognitive status of people with FH, it was difficult to ascertain whether there was any FH-specific neuropsychological impairment [69]. There is evidence to suggest that domains such as memory and executive functioning, as found in the present study, are some of the cognitive domains that are increasingly reported to be affected among people with FH [3,10,70]. Concerted efforts are required to establish expert consensus regarding FH disease-specific neuropsychological profiles, as has been previously established with other disorders that affect higher functioning [71,72].

Health literacy (HL) is increasingly recognized as an index that sheds light on the integrity of intellectual and psychological functioning among those with chronic diseases [21,24]. According to the U.S. Department of Health and Human Services, “health literacy is the degree to which individuals can obtain, process, and understand basic health information needed to make appropriate health decisions” [73]. For the present study, 3 items tapped into one’s subjective feelings about their ability to fill out medical forms, their dependency on assistance to help read hospital materials, and their problems learning about their medical conditions due to difficulty in understanding written information. According to the established criteria, the composite scores are grouped into ‘adequate’ or ‘inadequate’ HL [22,23]. In this study, approximately 34.6% of the FH patients endorsed themselves as having inadequate HL, and this appears to be orthogonal with the level of their education. None of the participants were illiterate. While is not clear whether HL is related to the natural process of a disease or something intimately linked to the ongoing treatment, many studies from different clinical populations have implicated variations of HL and cognition in the outcomes of the disease in question [24,25,26]. Studies have indicated that low HL is common among those from cross-cultural populations with chronic disease [74]. It is possible that obtaining, processing, and understanding basic health information might be related to the ‘sick role’ which, in turn, dictates the expected, patterned social role of being a dependent. In interdependent societies like those in the Arabian Gulf, when an individual succumbs to illness, the family takes over in order to safeguard the wellbeing of the sick person [75]. It is not clear whether such socially sanctioned behavior has the potential, in turn, to impede HL of each patient. To date, this is the first study on FH, HL, and cognition. Interestingly, the cognitive domain that was primarily associated with HL was attention and concentration. Intuitively, the diminution of these domains has the potential to render one incapable of sustaining attention or having intact working memory, both essential when deciphering basic health information [76]. More studies are needed to draw further conclusions from the present findings.

### Limitations

As is often the case in examining neuropsychological profiles, when there is no local normative data for the populations being studied, it is difficult to conclude whether the observed cognitive status constitutes ‘impairment’. More studies are needed to establish comparative data for cross-cultural populations, such as those in Oman. Secondly, some critics may perceive the present study as underpowered. It was not possible in the present ideographic approach to recruit a larger sample size, as is often done in traditional nomothetic studies [35]. However, it is also worthwhile to note that the present sample size appears to echo the sample sizes that have been used to previously explore the neuropsychological status of people with FH [3,7,10]. The third limitation is regarding the specificity of the cognitive domains, a pervasive feature of neuropsychological studies. For example, the present measures for current reasoning ability (Raven’s Progressive Matrices) is the traditional measure for Spearman’s ‘fluid intelligence’, with some recent data suggesting that this instrument has the potential to tap into working memory as well [77,78]. Similarly, the present operationalized executive functioning domains could also lack specificity [79]. This lack of specificity is intimately linked to existing neuropsychological batteries that have the potential to disallow the generalization of data. Thirdly, on one hand, given that there is a tendency to employ ‘bedside’ cognitive testing [8] in the currently existing literature, the advantage of this study is that it has employed properly standardized neuropsychological assessments. On the other hand, the missing link with the present data is the lack of brain imaging techniques that could confirm the presence of structural and functional changes in the brain. Fourth, the present study used a substantial number of questionnaires such as the WHO Quality of Life-BREF that are often prone to social-desirability bias. Fifth, it should be noted that there were four patients with homozygous FH. These patients do tend to have a more severe manifestation of the condition requiring more invasive treatment (lipid apheresis). It is therefore possible that their responses maybe be different from their heterozygous counterparts. However, since they are much smaller in number as compared to the heterozygous patients, we believe that the impact on the overall results will be quite minimal. Nevertheless, there is merit for future studies to explore whether heterozygous and homozygous FH patients show significant differences in performance across the various indices of biopsychosocial functioning. Finally, this study was conducted among patients in a single tertiary care center. The results obtained from the current sample may be affected by socio-economic or geographical factors that limit generalization. Therefore, community surveys or multicenter studies are warranted.

## 5. Conclusions

To date, there are only a few studies employing the biopsychological paradigm to investigate the FH population. Diseases such as FH that trigger multifactorial symptomology are best suited to be viewed through the lens of the biopsychosocial model. Among several diverse variables explored in the present study, it appears that the cognitive domain plays a central role in the neuropsychological expression of FH, manifesting primarily as issues with reasoning ability, attention and concentration, memory, and executive functioning. Issues with attention and concentration were also seemingly linked with inadequate health literacy. More studies are required to confirm the various pathways that lead to FH-specific neuropsychological disease expression. The present data appears to corroborate the view that chronic or life-limiting diseases, such as those that impact the integrity of the coronary artery, have the potential to compromise biopsychosocial functioning.

## Figures and Tables

**Figure 1 brainsci-12-01127-f001:**
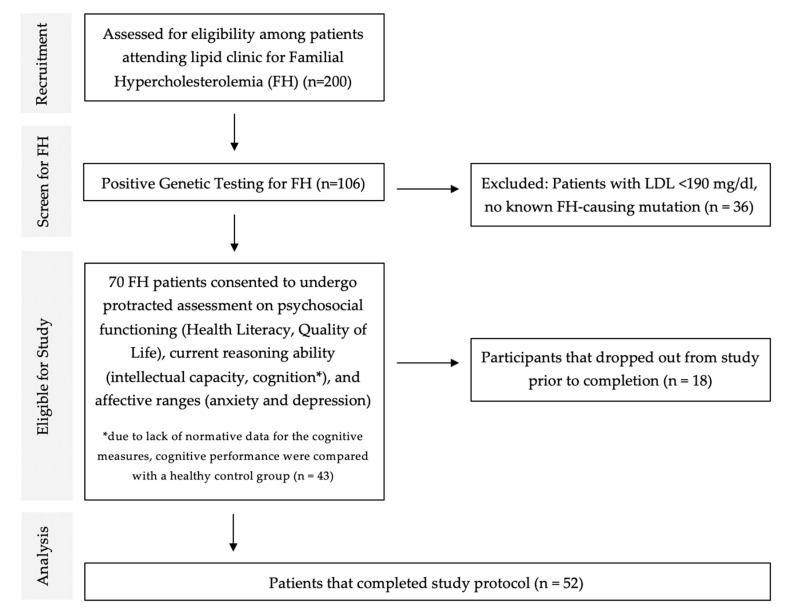
Flowchart indicating the process of participant recruitment for the current study.

**Table 1 brainsci-12-01127-t001:** Characteristics of the study sample.

Characteristics/Demographic		FH (N = 52)	Healthy Control (N = 43)	Statistics (*p*-Value)
Sex	Male	27 (51.9)	17 (39.5)	1.45 ^a^ (0.228)
	Female	25 (48.1)	26 (60.5)	
Marital status	Married	43 (82.7)	18 (41.9)	17.08 ^a^ (<0.001 **)
	Single	9 (17.3)	25 (58.1)	
Age (years)	Mean ± SD	37.2 ± 9.2	39.1 ± 5.8	1.25 ^b^ (0.216)
	Median [Range]	38.0 [21.0–52.0]	40.0 [29.0–56.0]	
Education level	High school/below	26 (50.0)	19 (44.2)	0.32 ^a^ (0.572)
	University/above	26 (50.0)	24 (55.8)	
Socio-economic status (Family income)	High	5 (9.6)	7 (16.3)	2.99 ^a^ (0.224)
	Middle	41 (78.8)	27 (62.8)	
	Low	6 (11.5)	9 (20.9)	
Cardiovascular disease	Yes	8 (15.4)	NA	
	No	44 (84.6)	NA	
Health Literacy (HL) *Health Communication Questionnaire*	Inadequate (≤9)	18 (34.6)	NA	
	Adequate	34 (65.4)	NA	
Quality of Life (QoL) *WHO Quality of Life-BREF*	Physical (Inadequate)	12 (23.1)	NA	
	Psychological (Inadequate)	10 (19.2)	NA	
	Social Relationships (Inadequate)	14 (26.9)	NA	
	Environmental (Inadequate)	4 (7.7)	NA	
	General health (Inadequate)	11 (21.2)	NA	
	General QoL (Inadequate)	7 (13.5)	NA	

FH: Familial Hypercholesterolemia; QoL: WHOQOL-BREF (Inadequate≤ 60, Adequate >60); HL, Health Communication Questionnaire (3 items); a: *t*-test; b: χ^2^ test; **: Sig. at *p* < 0.01; NA, not available.

**Table 2 brainsci-12-01127-t002:** Univariate analysis to compare the Intellectual capacity, cognitive and affective functioning outcomes between the FH patients (n = 52) and healthy controls (n = 43).

Intellectual and Cognitive Functioning		FH (N = 52)	Healthy Control (N = 43)	Univariate
		n (%)	n (%)	Statistics (*p*-Value)
Current Reasoning Ability (Raven’s Progressive Matrices)	Mean ± SD	31.8 ± 10.8	36.7 ± 8.9	2.37 ^a^ (0.020 *)
	Median [Range]	25.0 [24.0–75.0]	36.0 [25.0–75.0]	
Attention and Concentration Digit Span	Mean ± SD	16.8 ± 2.6	18.8 ± 2.5	3.94 ^a^ (<0.001 **)
	Median [Range]	18.0 [12.0–23.0]	18.0 [12.0–24.0]	
Memory (Long Delay Free Recall) California Verbal Learning Test	Mean ± SD	8.9 ± 2.5	11.4 ± 3.1	4.52 ^a^ (<0.001 **)
	Median [Range]	9.0 [5.0–15.0]	11.0 [4.0–20.0]	
Executive Functioning Verbal Fluency Test	Mean ± SD	16.0 ± 4.7	19.2 ± 2.5	4.23 ^a^ (<0.001 **)
	Median [Range]	16.0 [5.0–27.0]	19.0 [15.0–25.0]	
Executive Functioning Trail Making Test	Mean ± SD	217.7 ± 70.5	97.0 ± 24.1	11.45 ^a^ (<0.001 **)
	Median [Range]	240.0 [79.0–301.0]	92.5 [76.0–235.0]	
Hospital Anxiety and Depression Scale-Anxiety	Yes (≥8) No	17 (32.7) 35 (67.3)	23 (53.5) 20 (46.5)	4.18 ^b^ (0.041 *)
Hospital Anxiety and Depression Scale-Depression	Yes (≥8) No	13 (25.0) 39 (75.0)	8 (18.6) 35 (81.4)	0.56 ^b^ (0.455)

FH: Familial Hypercholesterolemia; IQ: Raven’s progressive matrices score; Attention: Wechsler Adult Intelligence Scale; CVLT: California Verbal Learning Test; VF: Verbal Fluency Test; TMT: Trail Making Test; HADS: Hospital Anxiety and Depression scale; a: *t*-test; b, χ^2^ test; *, Sig. at *p* < 0.05; **, Sig. at *p* < 0.01.

**Table 3 brainsci-12-01127-t003:** Univariate analysis to compare the cognitive functioning with indices of health literacy status.

Intellectual and Cognitive Functioning				Univariate
		Inadequate HL (N = 18)	Adequate HL (N = 34)	Statistics ^a^ (*p*-Value)
Current Reasoning (Raven’s Progressive Matrices)	Mean ± SD	32.8 ± 9.6	31.3 ± 11.5	244.50 (0.196)
	Median [Range]	30.0 [25.0–60.0]	25.0 [24.0–75.0]	
Attention and Concentration Digit Span	Mean ± SD	15.6 ± 2.7	17.4 ± 2.4	186.00 (0.017 *)
	Median [Range]	16.0 [12.0–21.0]	18.0 [12.0–23.0]	
Memory (Long Delay Free Recall) California Verbal Learning Test	Mean ± SD	8.6 ± 2.3	9.0 ± 2.6	271.00 (0.495)
	Median [Range]	8.0 [6.0–14.0]	9.0 [5.0–15.0]	
Executive Functioning Verbal Fluency Test	Mean ± SD	15.2 ± 4.2	16.4 ± 4.9	252.50 (0.301)
	Median [Range]	15.0 [7.0–21.0]	16.5 [5.0–27.0]	
Executive Functioning Trail Making Test	Mean ± SD	215.2 ± 75.4	218.9 ± 69.0	277.50 (0.818)
	Median [Range]	240 [102.0–301.0]	238.0 [79.0–301.0]	

HL: Health Literacy, Health Communication Questionnaire (3 items), ≤9 inadequate HL; a: Mann–Whitney U test; *, Sig. at *p* < 0.05.

## Data Availability

The data presented in this study are available on request from the corresponding author. The data are not publicly available due to ethical reasons.

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
