# Peer review of "A Comprehensive Neuropsychological Study of Familial Hypercholesterolemia and Its Relationship with Psychosocial Functioning: A Biopsychosocial Approach"

_brainsci, 2022, doi:10.3390/brainsci12091127_

Round 1

Reviewer 1 Report

In genotypically defined FH patients from Oman, cognitive tests and surveys were administered to investigate the relationship between FH, health literacy and cognitive function.

Methods

The health literacy study in FH administered across several countries was referred to but the reference was not provided, please add:

Hagger MS, Hardcastle SJ, Hu M, Kwok S, Lin J, Nawawi HM, Pang J, Santos RD, Soran H, Su TC, Tomlinson B. Health literacy in familial hypercholesterolemia: A cross-national study. European journal of preventive cardiology. 2018 Jun 1;25(9):936-43.

Results

Of those that refused the study, were their characteristics different from the cohort that completed the study?

Are clinical characteristics of the control group (other than FH status) matched? Please report the control data. The control group needs to match, this is an essential part of the scientific method.

Study flow diagram should be presented before the study results

Discussion

Please add this paper to the discussion:

Chang NT, Su TC. Investigating the association between familial hypercholesterolemia and perceived depression. Atherosclerosis Supplements. 2019 Mar 1;36:31-6.

Author Response

  1. Methods
    The health literacy study in FH administered across several countries was referred to but the reference was not provided, please add:
    Hagger MS, Hardcastle SJ, Hu M, Kwok S, Lin J, Nawawi HM, Pang J, Santos RD, Soran H, Su TC, Tomlinson B. Health literacy in familial hypercholesterolemia: A cross-national study. European journal of preventive cardiology. 2018 Jun 1;25(9):936-43.
    Author Response: 
    Thank you for these comments and suggestions.  The citation has been inserted into the text and reference section. 
  2. Results
    Of those that refused the study, were their characteristics different from the cohort that completed the study?
    Author Response: 
    Thank you for raising this issue. We have addressed it and now we have highlighted it in the text.

  3. Are clinical characteristics of the control group (other than FH status) matched? Please report the control data. The control group needs to match, this is an essential part of the scientific method.
    Author Response: 
    Thank you for raising this important issue. We have now clarified this issue along these lines:  The  patients were socio-demographically matched with with the norma,l healthy control group. (see Table 1)

  4. Study flow diagram should be presented before the study results
    Author Response: 
    Thank you. This has been done as suggested.

  5. Discussion

    Please add this paper to the discussion:

    Chang NT, Su TC. Investigating the association between familial hypercholesterolemia and perceived depression. Atherosclerosis Supplements. 2019 Mar 1;36:31-6
    Author Response: 
    Thank you for this suggestion.  This has been done.

Reviewer 2 Report

Minor comments:

The authors mention a rate of 32% anxiety and 25% depression (line 46-47) - how are those rates in the  general population, in Oman and other countries? I suggest also discussing this briefly in the discussion section (eg. discussion section line 416-417).

Abstract, Conclusion Section, sentence 1 (lines 52-53) is introduction/background and should be removed. Please specify what the "core features" (line 54) are.

How are the results compared to other chronic diseases? Are those results expected or usual in hereditary, chronic illnesses? Are there any particularities that might be worth mentioning in FH patients (e.g. due of the higher risk of CVD?)?

Major comments:

Did the authors include heterozygous as well as homozygous FH patients? How many were homozygous?

Did patients receive lipid apheresis? If yes, how many? Did these patients perform differently to non-lipid apheresis patients? Since lipid apheresis is an invasive, time consuming procedure it could be expected that it is also psychologically more strenuous and has an impact on the assessed parameters and psychosocial functioning, compared to patients on oral medication only. Also, disease burden may be more severe and thus impact the assessments.
If homozygous or patients on lipid apheresis were included, I suggest that these groups should be analyzed seperately.

Table 1 should also display values for healthy individuals / the control group, since this information is crucial in order to adequately interpret the results. How are the mentioned characteristics for the control group (education, socio-economic status, CVD, HL etc)? Please mention the observed differences between groups and discuss them. How might  they impact the questionnaires/assessments?

Author Response

  1. The authors mention a rate of 32% anxiety and 25% depression (line 46-47) - how are those rates in the  general population, in Oman and other countries? I suggest also discussing this briefly in the discussion section (eg. discussion section line 416-417).
    Author Response:
    Our apologies as this description was a bit convoluted. We were reporting the present data as shown in Table 1. The text has been made more succinct and the context changed.

  2. Abstract, Conclusion Section, sentence 1 (lines 52-53) is introduction/background and should be removed. Please specify what the "core features" (line 54) are.
    Author Response:
    Thank you for raising this suggestion.  We agree and the text has been amended accordingly. 

  3. How are the results compared to other chronic diseases? Are those results expected or usual in hereditary, chronic illnesses? Are there any particularities that might be worth mentioning in FH patients (e.g. due of the higher risk of CVD?)?
    Author Response:
    Thank you for these comments. We agree that the present manuscript needs such emphasis. We have done so in both the abstract and conclusion.  This issue was further recapitulated in the Discussion section.

  4. Did the authors include heterozygous as well as homozygous FH patients? How many were homozygous?
    Author Response:
    Thank you for this important query. Yes, both homozygous and heterozygous FH patients were included in this study. A total of four patients were classified as being homozygous. This information has now been included in the ‘Sample Characteristics’ sub-section of the Results.

  5.  Did patients receive lipid apheresis? If yes, how many? Did these patients perform differently to non-lipid apheresis patients? Since lipid apheresis is an invasive, time-consuming procedure it could be expected that it is also psychologically more strenuous and has an impact on the assessed parameters and psychosocial functioning, compared to patients on oral medication only. Also, disease burden may be more severe and thus impact the assessments.
    If homozygous or patients on lipid apheresis were included, I suggest that these groups should be analyzed seperately.

    Author Response:
    Thank you very much to the reviewer for these highly valuable questions.

    The four homozygous patients received LDL apheresis. The LDL cholesterol of these patients were higher than the patients with heterozygous FH. And even with the LDL apheresis, they are all experience coronary artery disease. Since the number of homozygous FH is very small compared to the number of heterozygous patients included in the study, we believe that it would have a major impact on the final study findings.

    These concerns have now been addressed as part of the limitations

  6. Table 1 should also display values for healthy individuals / the control group, since this information is crucial in order to adequately interpret the results. How are the mentioned characteristics for the control group (education, socio-economic status, CVD, HL etc)? Please mention the observed differences between groups and discuss them. How might  they impact the questionnaires/assessments?
    Author Response:
    Thank you for this suggestion. A simple comparison between the two groups (FH vs. healthy controls) on demographic factors was included in Table 1.